# Health-Related Quality of Life after Fractures of the Distal Forearm in Children and Adolescents—Results from a Center in Switzerland in 432 Patients

**DOI:** 10.3390/children9101487

**Published:** 2022-09-28

**Authors:** Thoralf Randolph Liebs, Alex Lorance, Steffen Michael Berger, Nadine Kaiser, Kai Ziebarth

**Affiliations:** Inselspital, Department of Paediatric Surgery, University of Bern, 3010 Bern, Switzerland

**Keywords:** fracture, forearm, radius, ulna, surgery, conservative treatment, health-related quality of life

## Abstract

(1) Background: We aimed to evaluate the health-related quality of life (HRQoL) in children with fractures of the distal forearm and to assess if HRQoL was associated with fracture classification; (2) Methods: We followed up on 432 patients (185 girls, 247 boys) who sustained a fracture of the distal radius or forearm from 1/2007 to 6/2007, 1/2014 to 6/2014, and 11/2016 to 10/2017. Patients filled in the Quick-DASH (primary outcome) and the Peds-QL; (3) Results: The radius was fractured in 429 and the ulna in 175 cases. The most frequent injury of the radius was a buckle fracture (51%, mean age 8.5 years), followed by a complete metaphyseal fracture (22%, 9.5 years), Salter-Harris-2 fracture (14%, 11.4 years), greenstick fracture (10%, 9.3 years), Salter-Harris-1 fracture (1%, 12.6 years), and other rare injuries. The most common treatment was closed reduction and an above-elbow cast in 138 cases (32%), followed by a cast without reduction (30%), splint (28%), and K-wire fixation and cast (9%). Definite treatment was performed initially in 95.8%, a new cast or cast wedging was performed in 1.6%, and revision surgery was performed in 2.5%. There were no open reductions and no plate fixations. After a mean follow-up of 4.2 years, patients with buckle fractures had a mean Quick-DASH of 3.3 (scale of 0–100) (complete fracture: 1.5; greenstick: 1.5; SH-1: 0.9; SH-2: 4.1; others: 0.9). The mean function score of the PedsQL ranged from 93.0 for SH-2 fractures to 97.9 for complete fractures; (4) Conclusions: In this cohort of 432 children with fractures of the distal forearm, there was equally good mean mid- and long-term HRQoL when assessed by the Quick-DASH and the PedsQL. There was a trend for children with complete metaphyseal fractures reporting better HRQoL than patients with buckle fractures or patients with Salter-Harris II fractures, however, these differences were not statistically significant nor clinically relevant.

## 1. Introduction

In children, fractures of the distal forearm are remarkably common. In most cases, they are the result of a fall on the outstretched hand. These fractures include buckle fractures, greenstick fractures, complete metaphyseal fractures, and fractures involving the growth plate. The latter are commonly classified according to Salter and Harris.

It appears accepted, that non-displaced fractures are treated with a cast, while a long-arm cast is traditionally used for unstable fractures, and a short-arm cast is used for stable fractures [1].

The treatment of displaced distal metaphyseal fractures is less clear. Several aspects have to be taken into account when treating these fractures in children who have open growth plates. These factors include—but are not limited to—the classification of the fracture, the age of the patient, the amount of dislocation in the frontal or lateral plane, the expected remaining growth potential [1], concomitant injuries, and expectations from both the children and the parents. As this list contains factors such as the age of the patient and the remaining growth potential, it is easy to understand that treatment decisions for these fractures are far more complex than in adults, where these parameters are not applicable. Therefore, it is easy to understand, that currently there are no evidence based recommendations regarding the treatment of these injuries depending on patient age, remaining growth, fracture classification and fracture dislocation [2].

Many authors report good results when reducing these fractures with some sort of sedation and/or analgesia and using fixation with a long-arm cast. Some authors recommend an additional K-wire to maintain a reduction in the cast [1]. Even other authors report the use of open reduction and fixation with plates [3,4].

Salter-Harris II fractures are of particular concern as these comprise the majority of physical injuries [5].

In paediatric orthopaedics, traditionally outcome assessment is performed radiographically. For instance, good results are reported when the radiographs demonstrate a good position. However, radiographs might not necessarily capture the subjective outcome assessment of the child. In order to capture the treatment result from the patient’s perspective, patient-reported outcome measures (PROMs) have been increasingly used recently. It has been stated that “the ultimate goal of health care is to restore or preserve functioning and well-being related to health, that is health-related quality of life” [6].

There is a lack of studies assessing PROMs in children after they have sustained a fracture of the distal forearm. It is also unknown if the health-related quality of life (HRQoL) differs by fracture classification, fracture severity, or treatment performed.

Therefore, we have initiated this study to evaluate the HRQoL in children who have sustained a fracture of the distal forearm. In addition, we evaluated if HRQoL was associated with fracture classification, fixation method, secondary displacement, or revision surgery.

## 2. Materials and Methods

This is a retrospective analysis, in which patients who were treated for a distal fracture of the forearm were contacted by postal mail. The regional ethics committees gave their approval to the study protocol (both the ethcis committees of the Paediatric Clinics of Inselspital, University of Bern, and the Ethics Commission of the Canton of Bern).

There are some methodological similarities to sister studies in which the health-related quality of life (HRQoL) after fractures of the femur [7], lateral third of the clavicle [8], fractures of the proximal humerus [9], or supracondylar fractures of the humerus [10] in children and adolescents were assessed.

### 2.1. Patients

All sequential patients with an age of up to 16 years, who sustained a fracture of the distal radius or forearm from January to June 2007, January to June 2014, and November 2016 to October 2017 and who were treated at our institution were candidates for inclusion in the study. Serving more than a million people, our facility is among the leading pediatric trauma centers in Switzerland.

Patients were identified on the basis of radiological reports within our Picture Archiving and Communication System.

Exclusion criteria were: (1) other significant trauma requiring therapy, (2) initial treatment performed outside our institution, (3) incapacity to complete the questionnaires because of the language barrier or cognitive limitations (Figure 1).

### 2.2. Radiological Analysis

Initially, all images were assessed by paediatric radiologists. As a second step, all fractures were categorized using the radiological AO classification system [11]. Interobserver bias was prevented by having one of the authors (A.L.), who did not know the patient’s clinical outcome, perform this step. Radiographs were also assessed regarding growth arrest. If that author had doubts about his assessment, he contacted the principal author. In all these cases, the doubts could be clarified. Since the author who classified the images gained more and more experience in assessing the images, he reclassified all images a second time, thereby limiting the probability of intraobserver error.

### 2.3. Data Collection

Starting in February 2018, we mailed study-related information, a consent form, and questionnaires to the participants (or their parents, depending on their age at the time) (Figure 1). Participants who did not reply received three postal-mail-based reminders. To find out why they were not replying, participants who were not yet responding were phoned. At that time, an attempt was made to administer the survey over the phone [10].

As in the sister studies [8,9,10], we used the disease-specific Quick-DASH (Disabilities of the Arm, Shoulder and Hand) in a validated translation [12,13] as the primary outcome measure. Responses were recorded on a five point Likert scale. Scores were standardised to 0–100, with higher scores indicating more disability. If more than 10% of the items were unanswered, a Quick-DASH score was regarded as missing.

We have chosen the validated translated version of the Paediatric Quality of Life Inventory (PedsQL) as a secondary outcome [14]. Higher scores indicated more physical or social function, and scores ranged from 0–100.

Radiographs and the patient’s chart were used to gather information on the patient’s demographics, the dates of the injury, the side (right/left), and the chosen treatment. We included questions about handedness and concurrent injuries in the survey [8,9,10].

We were able to follow up on 432 patients (185 girls, 247 boys) who sustained a fracture of the distal radius or forearm from January to June 2007, January to June 2014, and November 2016 to October 2017, at an average age of 9.3 (SD 3.7) years.

### 2.4. Treatment Algorithm

#### 2.4.1. Initial Treatment


**Non reduction and casting:**


If the fracture was not dislocated, we used casting only.


**Reduction and casting:**


If the fracture was dislocated and the fracture was stable after reduction, we applied an upper-arm cast.


**CRPP:**


If the fracture was dislocated and the fracture was not stable after reduction, we reduced the fracture and used a percutaneous (unburied) retrograde applied K-wire that was introduced at the tip of the styloid process of the radius and exited the radius proximal of the fracture through the cortex (bicortical fixation). Usually, only one wire was used in order to keep the trauma to the growth-plate to a minimum. A cast was then applied.

If there is adequate pain control and no compromise in perfusion or nerve function, we avoid to perform surgery after midnight and postpone it to the next day [15].

#### 2.4.2. Further Treatment

If the fracture was reduced but no K-wire fixation was used, we perform a radiographic follow-up after 5–7 days in order to exclude a secondary dislocation within the cast.

The cast is usually applied for a period of 4 weeks in patients younger than 10 years and for 5 weeks in patients older than 10 years.

After that time a clinical and radiological follow-up is performed and it is decided if the cast treatment can be discontinued. If a K-wire has been used, typically it can be removed during an outpatient procedure using nitrous oxide and/or nasally applied fentanyl. If the range of motion is restricted, patients are invited back for another appointment after four weeks. Physiotherapy is only taken into consideration at that time. As—in our experience—these cases are rare, they do not justify routine physiotherapy after these common fractures [16].

Neurapraxias are monitored until they resolve spontaneously. Since transient traumatic neurapraxia is frequently noted in this population, these were not considered as complications [10].

### 2.5. Statistical Analysis

First, we analysed the outcome measures, such as Quick-DASH and PedsQL. Then, we looked for associations between HRQoL and radiological fracture type.

In our dataset, most outcome measures demonstrated a clear ceiling effect—that is, most cases were clustered close to the best conceivable outcome. Such distributions can not be adequately visualized using standard box plots. For this reason, we have selected violin plots instead. In violin plots, the width of the graph represents the probability density of the data (comparable to a mirrored histogram rotated by 90 degrees), allowing a suitable graphical representation [10].

All *p*-values are two-tailed. We did not perform corrections for multiple comparisons. Statistical analysis was performed using Statistical Package for the Social Sciences (SPSS Inc., Chicago, IL, USA) and R [17].

## 3. Results

The radius was fractured in 429 cases and the ulna in 175 cases. Both bones were fractured in 173 cases. The most frequent injury of the radius was a buckle fracture (222 children, 51 percent, mean age 8.5 years), followed by a complete metaphyseal fracture (93 children, 22 percent, 9.5 years), Salter-Harris type 2 fracture (62 children, 14 percent, 11.4 years), greenstick fracture (42 children, 10 percent, 9.3 years), Salter-Harris type 1 fracture (5 children, 1%, 12.6 years), and other less frequent injuries (Salter-Harris type 4 (n = 1), Peterson type 1 (n = 2), complex (n = 2)).

The most common treatment was a closed reduction and the application of an above elbow cast in 138 cases (32 percent), followed by a cast without prior reduction (129 children, 30 percent), splint (119 children, 28 percent), and closed reduction with K-wire fixation and above elbow cast (39 cases, 9 percent). Definite treatment was performed initially in 95.8 percent of children, a modification (new cast or cast wedging) was performed in 7 cases (1.6 percent), and revision surgery was performed in 11 cases (2.5 percent). There were no open reductions and there was no plate osteosynthesis (Table 1).

After a mean follow-up of 4.2 years, patients with a buckle fracture had a mean Quick-DASH of 3.3, at a scale of 0–100, with lower values representing better HRQoL (complete fracture: 1.5; greenstick: 1.5; Salter-Harris type 1: 0.9; Salter-Harris type 2: 4.1; others: 0.9). The mean function score of the PedsQL ranged from 93.0 for SH-2 fractures to 97.9 for complete fractures, at a scale of 0–100, with higher values representing better HRQoL (Table 2, Figure 2).

A graphical presentation of the physical and the social function score of the Peds-QL is given in Figure 3 and Figure 4.

There was no statistically significant difference in the association between HRQoL and AO radiological classification or type of surgical treatment. In addition, there were no statistically significant associations between HRQoL and the need for revision surgery in the univariate analysis (Figure 5).

There were no complications requiring interventions, revision surgery, or manipulation. There was no growth arrest.

## 4. Discussion

This study clearly showed that the treatment protocol described in this study for children with a fracture of the distal forearm is associated with good health-related quality of life (HRQoL) as measured with the Quick-DASH and the Peds-QL at a mean of 4.2 years follow-up. Our analysis of this injury is one of the largest in the literature and one of the few assessing the HRQoL in this population. These excellent results were independent of radiological fracture type, or treatment performed.

### 4.1. Health-Related Quality of Life and Radiological Fracture Type

The main focus of our study was to report the HRQoL of our population who were treated according to our treatment regime. As could be shown, the results are—on the whole—remarkably good with only minor differences between study groups.

As the most common injury of the radius was a buckle fracture (51%), which is considered to heal without any sequela, we initially considered to exclude this group in our study. However, we reckoned that this group might serve as an appropriate control group since we expected worse HRQoL in the children who sustained other fracture types. To our surprise, patients with buckle fractures had a mean Quick-DASH of 3.3 on a scale of 0–100, which was worse than the HRQoL of the complete metaphyseal fractures (Quick-DASH 1.5), the greenstick fractures (1.5), or the Salter-Harris-1 fractures (0.9). Only the Salter-Harris-2 fracture had a slightly worse HRQoL (4.1) when compared to the buckle fractures. However, it must be noted that the differences are slight and should therefore not be overinterpreted. Further studies are needed to assess if these non-perfect numbers for buckle fractures can be indeed attributed to the fracture, or if it is just random noise that is inherent to the recording of the outcome measure.

If only a cast fixation has been chosen there is a risk of a secondary dislocation. For this reason, we always aim to follow up patients with a risk for a secondary dislocation at 5–7 days after cast application. As could be seen from the results, a modification (new cast or cast wedging) was performed in 7 cases (1.6 percent), and revision surgery was performed in 11 cases (2.5 percent). We, therefore, asked if the HRQoL was inferior in this patient group. As can be seen in Figure 5, there was a good HRQoL in the patients undergoing cast wedging and a slightly inferior HRQoL in the patients who have undergone revision surgery. Further analysis revealed that among that group, patients with a Salter-Harris 2 fracture had slightly worse scores; however, those scores were not statistically significant.

We tried to compare our results to the literature. Unfortunately, we were no able to find many other studies assessing outcome measures such as the Quick-DASH or the Peds-QL in patients who had undergone a fracture of the distal radius or forearm. Musters et al. reported DASH scores for 51 children who underwent either above or below elbow cast for mainly greenstick fractures of the distal forearm and reported DASH scores of 2.1 and 4.4. for these groups [18]. Peterlein et al. [19] reported a remarkably good mean DASH score of 0.4 in 90 patients who had undergone elastic-stable intramedullary nailing (ESIN) of diaphyseal forearm fractures in childhood fractures. In addition, there are two study protocols for fractures of the forearm shaft [20] or metaphyseal [21], but no results yet. Overall, our Quick-DASH is comparable to the few other studies published up to this time, although patient numbers in that studies are much smaller and different fractures were examined.

### 4.2. Limitations

Our observations must be interpreted in light of several limitations, which apply to the sister studies [7,8,9,10] as well: First, as this was a mono-centre study it could be argued that external validity is limited. However, a bias in the run-in phase is unlikely because we are the only hospital in a large geographic area addressing paediatric trauma and we included all consecutive cases, making a high external validity probable [8]. However, given that there is significant diversity in orthopaedic treatments across and within nations, we are aware that no single study is capable of giving full external validity [22]. Second, the radiographs under examination were not specially prepared for this analysis; they were just made routinely. Therefore, the setting of these radiographs is comparable to the situation of the clinician [8]. Since the individual classifying the fractures was unaware of the patient’s clinical outcome, the radiological assessment could be viewed as being blinded [8]. Third, due to its retrospective design, this study suffers from methodological weaknesses common in this design. This includes for example no intermediate data points and missing data on the HRQoL prior to the injury. Although the latter is viewed as a methodological flaw in research analyzing adult fractures, this does not always hold true for paediatric fractures because children typically do not have physical limits prior to the injury and we excluded children with prior or concurrent injuries. Therefore, it is reasonable to believe that limitations of the disease-specific outcome measure are in fact attributable to the injury [8]. Fourth, the Quick-DASH has not yet been formally validated in this age group and in some cases the parents filled in the questionnaire based on their assessments of their children’s functioning (by proxy). However, the Quick-DASH has been used by numerous authors for the evaluation of paediatric other upper extremity fractures before [23,24,25,26,27,28,29,30,31,32,33,34]. Consequently it appears, that the DASH/Quick-DASH is most widely utilized outcome measure for paediatric upper extremity fractures. Fifth, our follow-up rate was 80%, which is just the recommended rate for follow-ups. However, most other studies we are aware of have a similar or lower rate of follow-up, e.g., [25,35,36,37]. To our knowledge, no study that examined distal forearm fractures was able to evaluate the HRQoL in as many children as did our investigation (n = 432). Sixth, although we had a large sample size it is possible that we missed existing associations of HRQoL to fracture patterns or treatment chosen. Given the excellent results of the overall group, which even exhibits a ceiling effect, it is unlikely that any such association would become clinically relevant. Seventh, we analyzed children of different time periods, as we assumed there might be differences over time. However, that topic will be the subject of further analysis in the future.

## 5. Conclusions

In conclusion, we report that children who sustained a fracture of the distal radius or forearm and who were treated according to this protocol had an excellent health-related quality of life. These findings show that the treatment protocol followed in this study is clear, does not include open reductions or plate fixations, and is associated with excellent treatment outcomes for this frequent injury.

## Figures and Tables

**Figure 1 children-09-01487-f001:**
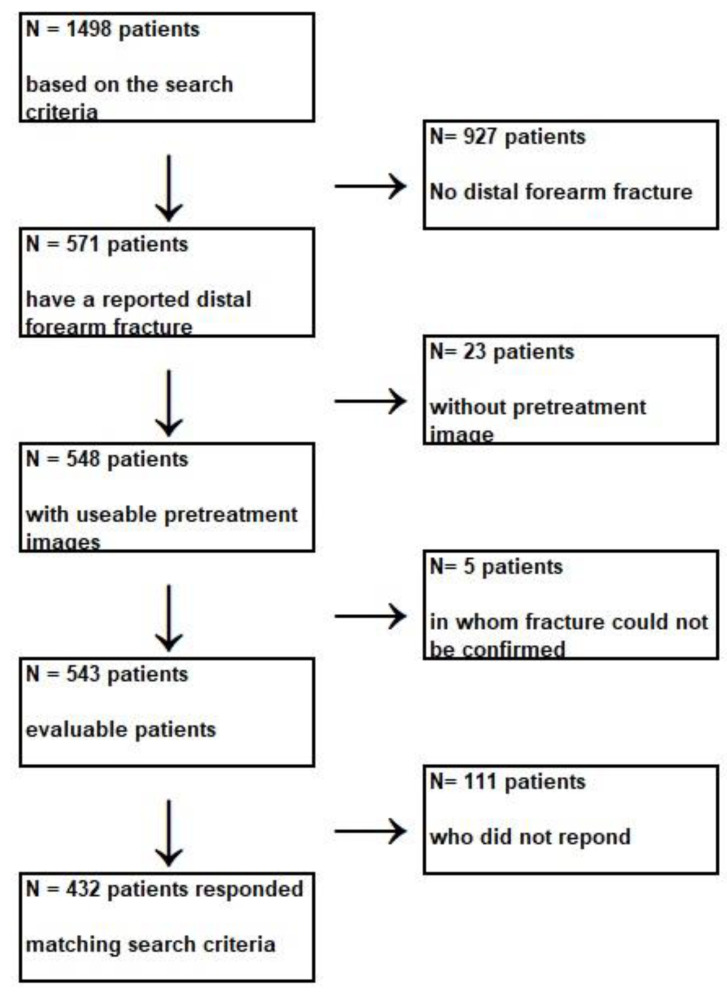
STROBE Participant Flow Chart.

**Figure 2 children-09-01487-f002:**
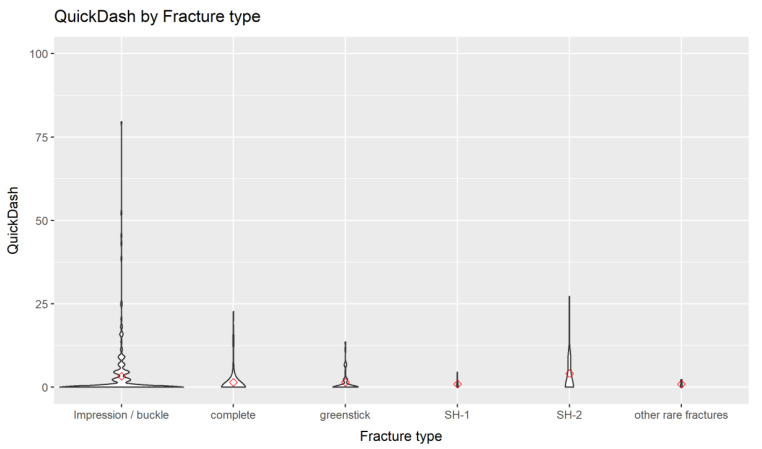
The primary outcome Quick-DASH at follow-up by fracture type of the radius. Since there were considerable ceiling effects, we used violin plots for the graphical presentation. Please see the methods section for a description of the violin plots.

**Figure 3 children-09-01487-f003:**
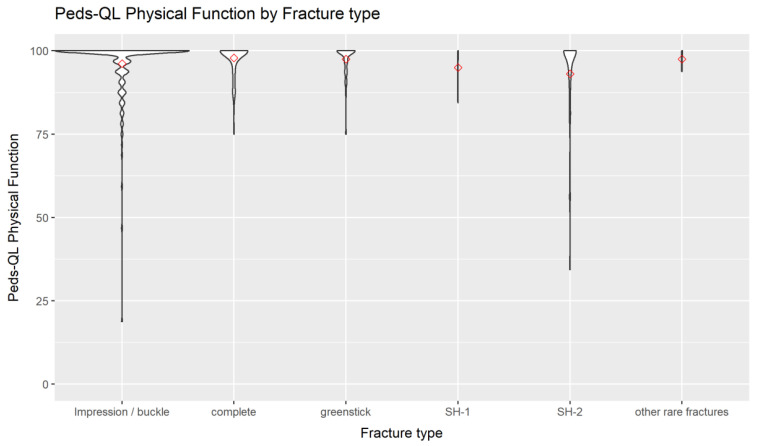
Peds-QL physical function at follow-up by fracture type of the radius.

**Figure 4 children-09-01487-f004:**
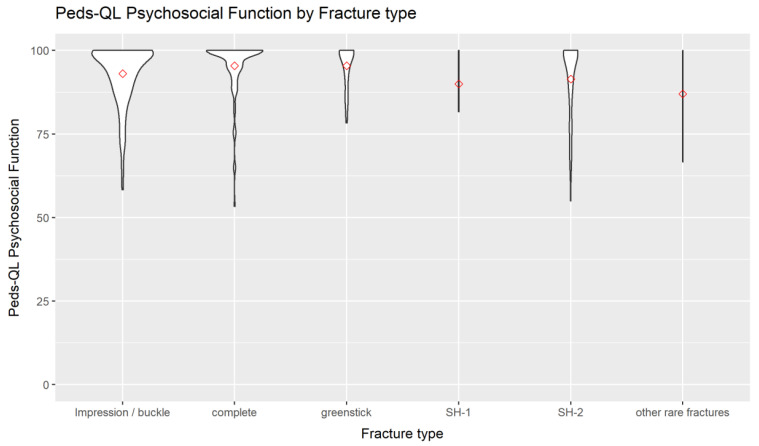
Peds-QL social function at follow-up by fracture type of the radius.

**Figure 5 children-09-01487-f005:**
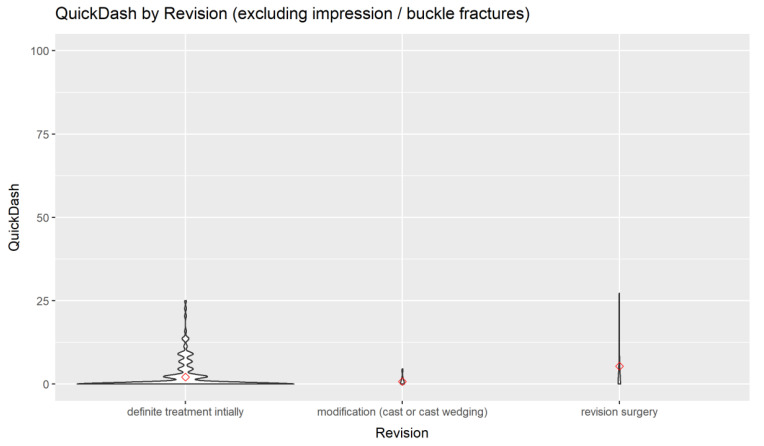
The primary outcome Quick-DASH at follow-up by revision. Please note that patients with impression/buckle fractures are excluded, since a revision is not performed in those fractures.

**Table 1 children-09-01487-t001:** Baseline characteristics.

		Type of Radius Fracture																																	
		Impression/Buckle					Compete Metaphyseal					Greenstick					SH1					SH2					Other Rare Fractures					Total			
		n	%	Mean	SD		n	%	Mean	SD		n	%	Mean	SD		n	%	Mean	SD		n	%	Mean	SD		n	%	Mean	SD		n	%	Mean	SD
gender	female	112	50%				31	33%				18	43%				1	20%				22	35%				1	20%				185	43%		
	male	110	50%				62	67%				24	57%				4	80%				40	65%				4	80%				244	57%		
Age at the time of injury [years]		222		8.52	3.87		93		9.45	3.36		42		9.29	2.97		5		12.55	1.62		62		11.24	3.07		5		9.95	3.64		429		9.25	3.68
Age at the time of injury	0 to <3 years	23	10%				1	1%				2	5%																			26	6%		
	3 to <6 years	36	16%				18	19%				3	7%									4	6%				1	20%				62	14%		
	6 to <9 years	62	28%				18	19%				14	33%									9	15%				1	20%				104	24%		
	9 to <12 years	51	23%				32	34%				15	36%				1	20%				23	37%				1	20%				123	29%		
	12 years and older	50	23%				24	26%				8	19%				4	80%				26	42%				2	40%				114	27%		
Follow-up duration [years]		222		3.93	4.07		93		4.72	4.63		42		3.79	4.24		5		7.13	6.25		62		4.28	4.44		5		5.47	5.93		429		4.18	4.31
Injured side (right vs. left)	right	82	37%				38	41%				22	52%				4	80%				26	42%				1	20%				173	41%		
	left	134	61%				54	58%				20	48%				1	20%				35	56%				4	80%				248	58%		
	both	4	2%				1	1%														1	2%									6	1%		
Injured side (dominat vs. non-dominant)	non-dominant side	138	63%				48	52%				23	55%				1	20%				42	68%				3	60%				255	60%		
	dominat side	82	37%				45	48%				19	45%				4	80%				20	32%				2	40%				172	40%		
Skin injury	No, the skin was intact	212	96%				87	94%				39	93%				4	80%				56	90%				5	100%				403	94%		
	Yes, there was a graze	9	4%				6	6%				2	5%				1	20%				4	6%									22	5%		
	Yes, a skin suture had to be done											1	2%									2	3%									3	1%		
Vessel injury	No, not that I know of	221	100%				92	99%				42	100%				5	100%				62	100%				5	100%				427	100%		
	Yes, but it was not necessary to suture a vessel or an artery	1	0%				1	1%																								2	0%		
	Yes, it was necessary to perform vascular sutures																																		
Numbness in fingers	No	180	81%				68	75%				34	81%				4	80%				37	60%				4	80%				327	77%		
	Yes, less than a week	37	17%				20	22%				8	19%									19	31%				1	20%				85	20%		
	Yes, for several weeks	4	2%				1	1%									1	20%				6	10%									12	3%		
	Yes, for more than 3 months	1	0%				1	1%																								2	0%		
	Yes, still ongoing						1	1%																								1	0%		

**Table 2 children-09-01487-t002:** Results by type of radius fracture.

		Type of Radius Fracture																																	
		Impression/Buckle					Complete Metaphyseal					Greenstick					SH1					SH2					Other Rare Fractures					Total			
		n	%	Mean	SD		n	%	Mean	SD		n	%	Mean	SD		n	%	Mean	SD		n	%	Mean	SD		n	%	Mean	SD		n	%	Mean	SD
Max. ROM for pronation	90 degrees	213	96%				85	91%				39	93%				5	100%				56	90%				4	80%				402	94%		
	45 degrees	8	4%				8	9%				3	7%									6	10%				1	20%				26	6%		
	0 degrees																																		
Max. ROM for supination	90 degrees	218	99%				87	94%				40	95%				5	100%				61	98%				5	100%				416	97%		
	45 degrees	2	1%				6	6%				2	5%									1	2%									11	3%		
	0 degrees	1	0%																													1	0%		
Abilty to throw a ball with the injured side	Yes, that is easily possible	209	95%				88	96%				42	100%				5	100%				61	98%				5	100%				410	96%		
	Yes, but just a bit	8	4%				4	4%														1	2%									13	3%		
	No, I am not able to	4	2%																													4	1%		
Impression that the forearm limits the force of the whole arm	not at all	186	84%				78	84%				36	86%				5	100%				39	63%				3	60%				347	81%		
	A little bit	31	14%				15	16%				5	12%									19	31%				1	20%				71	17%		
	moderate	2	1%									1	2%									3	5%				1	20%				7	2%		
	quite	3	1%																			1	2%									4	1%		
	very much																																		
Satisfaction with cosmetic result	very satisfied	205	94%				73	81%				38	90%				3	60%				45	74%				4	80%				368	87%		
	rather satisfied	11	5%				10	11%				3	7%				2	40%				11	18%				1	20%				38	9%		
	moderately satisfied	1	0%				5	6%				1	2%									3	5%									10	2%		
	A little bit satisfied	1	0%																			2	3%									3	1%		
	Not satisfied at all						2	2%																								2	0%		
Current treatment with pain killers	No	190	99%				84	100%				37	97%				3	100%				55	98%				5	100%				374	99%		
	Yes	2	1%									1	3%									1	2%									4	1%		
Point in time when forearm was used regulary	Immediately after treatment	5	2%				1	1%				1	2%																			7	2%		
	Immediately after removal of cast	130	59%				43	46%				15	37%				4	80%				26	42%				3	60%				221	52%		
	Other	66	30%				39	42%				17	41%									28	45%				2	40%				152	36%		
	Do not know	19	9%				10	11%				8	20%				1	20%				8	13%									46	11%		
Weeks after forearm was used regulary		222		4.44	2.96		93		8.00	5.72		42		5.59	4.32		5					62		6.38	3.19		5		5.00	4.24		429		5.86	4.26
Revision	definite treatment intially	221	100%				83	89%				42	100%				5	100%				55	89%				5	100%				411	96%		
	modification (cast or cast wedging)	1	0%				5	5%														1	2%									7	2%		
	revision surgery						5	5%														6	10%									11	3%		
Quick-DASH (0–100)		222		3.30	8.89		93		1.47	3.87		42		1.46	3.17		5		0.91	2.03		62		4.11	6.13		5		0.91	1.24		429		2.78	7.17
PedsQL, physical function (0–100)		222		96.16	9.01		93		97.88	4.57		42		97.47	4.89		5		95.00	7.19		62		93.04	13.03		5		97.50	3.42		429		96.21	8.70
PedsQL, psychosocial function (0–100)		222		93.04	8.92		93		95.38	8.50		42		95.42	6.35		5		90.00	7.91		62		91.45	11.51		5		87.00	15.29		429		93.44	9.20

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
