# Peer review of "Health-Related Quality of Life after Fractures of the Distal Forearm in Children and Adolescents—Results from a Center in Switzerland in 432 Patients"

_children, 2022, doi:10.3390/children9101487_

Round 1

Reviewer 1 Report

I think this is a good job. The authors make a correct exposition, as well as an accurate conclusion. However, the work is limited to one center, and it is rather a matter of corroborating whether the protocol adapted in that center can be correct or not. Perhaps the title should refer to this.

Reviewer 2 Report

I read with interest the manuscript entitled "Health-Related Quality of Life after Fractures of the Distal Forearm in Children and Adolescents"

I present suggestions and opinions below;

The word calcaneus is listed in the keywords. Please explain.

In the introduction, it is not necessary to repeat reference 1, but it is enough to mention it once at the end of the sentence.

After the sentence "As this list contains factors such as the age of the patient and the remaining growth potential, it is easy to understand that treatment decisions for these fractures are far more complex than in adults, where these parameters are not applicable." you definitely have to include certain references that talk about some factors, like angulation...for example: Bašković M. Acceptable angulation of forearm fractures in children [published online ahead of print, 2022 Jun 9]. Rev Esp Cir Orthop Traumatol. 2022;S1888-4415(22)00151-5. doi:10.1016/j.recot.2022.06.003

Do you think it is enough that the radiological images are assessed by one author? What did you do if the mentioned author had doubts about the assessment?

Have the radiological images been evaluated by a pediatric orthopedic surgeon or radiologist? Please write.

For the patients you contacted by phone to fill out the questionnaire, how did you ensure informed consent?

Also, as with reference 1, it is not necessary to repeat references 7-9 after each sentence.

You provided the following information: "The cast is usually applied for a period of 4 weeks in patients younger than 10 years and for 5 weeks in patients older than 10 years." Is this in line with the guidelines?

Why is physiotherapy only started after 4 weeks? Why not immediately after removing the immobilization, that is, removing the wires?

Please write how many surgeons participated in the treatment of your 432 patients. This is important information because it is known that the methods and principles of treatment differ from surgeon to surgeon.

In the results, I do not find information on how many cases the radius and ulna were broken at the same time.

When you talk about other less frequent injuries, please write in parentheses what all those injuries are.

In the results, you stated that there was no statistical significance between the groups, but you did not write p-values anywhere.

In the discussion section, limitations are usually written at the end of the section.

With regard to the time lag since the fracture, do you think there might be differences in the responses of children who broke their forearm 10 years ago, compared to children who suffered a fracture a year ago? Please explain.

In the discussion you mention figure 7, probably referring to figure 5.

It would be useful for the readers if they looked a little more at the possible complications in the treatment.

Round 2

Reviewer 2 Report

Please double-check all references and their proper citation in accordance with the instructions for authors.

for example;

·         .[3;4] -> [3,4].

·         [24;34-36] -> [24,34-36]

·         References should be written at the end of sentences.

·         Baskovic M. Acceptable angulation of forearm fractures in children. Rev Esp Cir Ortop Traumatol 2022 June 9. -> Bašković M. Acceptable angulation of forearm fractures in children. Rev. Esp. Cir. Ortop. Traumatol. 2022. Epub ahead of print.

etc...

Please see the following link;

Please, the explanation you gave to the question... "Do you think it is enough that the radiological images are assessed by one author?" ...add under the materials and methods section to make it clear to all readers how the images were evaluated. Also, mark all changes in the text in red to make it easier for reviewers to follow the changes.

If you deny that there are international guidelines, please write that you kept the immobilization in accordance with your own practice.

Please note the following reference;

Caruso, G., Caldari, E., Sturla, F.D. et al. Management of pediatric forearm fractures: what is the best therapeutic choice? A narrative review of the literature. Musculoskeletal Surg 105, 225–234 (2021). https://doi.org/10.1007/s12306-020-00684-6

Don't you think that physical therapy after four or more weeks is too late for some cases? Please support your answer with a literature reference.

Please, an opinion of the question...."With regard to the time lag since the fracture, do you think there might be differences in the responses of children who broke their forearm 10 years ago, compared to children who suffered a fracture a year ago?...add to the limitations.

Label the subsections adequately...eg. 2.1. Patients...

Please re-read and proofread the manuscript thoroughly, also ask a native English speaker to proofread the text.
